# Examination of hyper-palatable foods and their nutrient characteristics using globally crowdsourced data

Daiil Jun[1,2☯], Kelly Knowles[1,2☯], Tera L. Fazzino[1,2]*

1 Department of Psychology, University of Kansas, Lawrence, Kansas, United States of America, 2 Cofrin Logan Center for Addiction Research and Treatment, University of Kansas, Lawrence, Kansas, United States of America

☯ These authors contributed equally to this work and shared the first authorship.
* tfazzino@ku.edu

## Abstract

Hyper-palatable foods (HPF), characterized by specific nutrient combinations at moderate to high levels (e.g., fat and sodium), have been suggested to increase energy intake and obesity risk due to their strong reinforcing properties. The study examined patterns in HPF availability, nutrient characteristics, and overlap with ultra-processed foods (UPF) across countries from a globally crowdsourced, open-source dataset. Food data (N = 314229 food items) from 17 countries were analyzed. Compared to the US, foods from most European countries examined, as well as Canada and Mexico, were significantly less likely to be identified as HPF (ORs = 0.70 to 0.93) and had lower % of calories from fat, sugar, starchy carbohydrates, and sodium compared to HPF items from the US (ORs = 0.76 to 0.98). Across countries, items identified as HPF substantially exceeded HPF threshold criteria by 70–229%. Foods identified as being both HPF and ultra-processed foods ranged from 33% (Bulgaria) to 50% (US). Overall, findings from 17 countries across Europe, North America, and South America highlighted foods from the US as being more likely to be hyper-palatable relative to most other countries examined. Results highlighted variability in the degree to which foods met criteria as HFP and UPF across countries.

## Introduction

Evidence from neurobiological and behavioral studies has suggested that combinations of nutrients (e.g., fat and sodium; fat and sugar) in foods commonly found in the modern food environment may substantially enhance the reinforcing properties of such foods [1,2], increasing risk for excess intake and obesity [3,4]. Nutrient combinations present at moderate to high thresholds are hypothesized to be a mechanism that elevates the reinforcing properties of foods, increasing wanting or drive to obtain a (food) reward, and strengthening the value of a (food) reward for individuals. Thus,

**Data availability statement:** The detailed R code used for the data processing and the data files are publicly available online: https://osf.io/9juwz/

**Funding:** Grants from the Kansas Idea Network of Biomedical Research Excellence (P20 GM103418; PI: Wright) and the University of Kansas Center for Undergraduate Research supported author KK's time during the study. The funders had no role in study design, data collection and analysis, decision to publish, or preparation of the manuscript.

**Competing interests:** The authors have declared that no competing interests exist.

foods that contain combinations of nutrients (fat, sugar, sodium, and/or carbohydrates) at moderate to high levels are termed hyper-palatable foods (HPF; [5]).

HPF may have detrimental effects on our food seeking and intake behavior due to their strong reinforcing properties. HPF may produce an acutely rewarding eating experience, which may drive greater energy intake within an eating occasion [6], and repeated HPF intake over time may yield neurobiological changes that enhance wanting and drive to consume HPF (termed sensitization) [7]. Notably, wanting as driven by HPF is considered conceptually and neurobiologically distinct from food liking [7,8], which is the domain of sensory scientists and viewed as a subjective measure of food palatability [9,10]. Although food liking and palatability may contribute to food preferences and choice, food wanting and motivation have been well-established as strong behavioral drivers of excess intake and most reliably associated with obesity risk [2,11,12]. Therefore, it is necessary to understand HPF availability and characteristics in the global food environment, to contextualize public health risk.

Most of the literature characterizing HFP availability has focused on the food environment in the United States (US); little is known about HPF in the global food supply. Findings from the US are concerning and have indicated that HPF saturate the US food supply. As of 2018, HPF comprised 69% of available foods in the US food environment [13]. Beyond the US, one study characterized the prevalence of HPF in the Italian food environment and revealed that HPF comprised a minority (28.8%) of foods in the food supply [14]. Thus, wide variability in HPF availability was documented across two countries (US and Italy); however the presence of HPF in the global food supply has not been comprehensively characterized. Given that HPF may skew food seeking and intake behavior toward HPF and may yield substantial health consequences, it is critical to understand the characteristics and degree to which HPF are present in the global food supply, to form the basis for food environment policy interventions. Despite the lack of quantitative metrics on HPF availability, concern that HPF are prominent in food environments globally is well-founded. Multi-national food companies that originated in the US played a key role in disseminating HPF into the US food supply [15] and these companies have invested billions of dollars since the 1980's to penetrate food environments globally, with prominent footholds in countries in Europe and South America [16,17]. Thus, it is likely that multi-national companies have disseminated HPF into the European and South American markets, although research is needed characterize the extent and nature that HPF may be present.

In a related area of the literature to HPF, researchers have more robustly characterized the availability of ultra-processed foods, as defined by Monteiro and colleagues [18,19] using the NOVA classification system. Ultra-processed foods (UPF) are identified based on the nature and extent of industrial processing, and are considered to contain little whole food contents and are designed for convenience [18,19]. Globally, UPF account for 50% or more of daily energy intake in the US [20] and the United Kingdom [20], and 20% to 40% of that in Brazil [21], Japan [22], Mexico [23], and Portugal [24]. Global UPF sales have shown a consistent increase worldwide, especially rapidly in Asia, the Middle East, and Africa [25].

UPF may contain combinations of palatability-related nutrients and meet criteria as HPF; however UPF and HPF are distinct conceptually, in their mechanisms that may yield obesity risk, and in the foods they identify in the food supply. Specifically, HPF are identified using quantitative criteria and have an explicit mechanism hypothesized to yield their reinforcing properties: nutrient combinations at thresholds not found in nature. Due to the quantitative criteria, the HPF definition can be applied to all foods. UPF are defined descriptively and are identified by reviewing the ingredients list of foods. The UPF definition itself does not specify a mechanism that may yield their health consequences; however researchers have posited multiple mechanisms including the presence of refined ingredients (e.g., refined carbohydrates, added sugars) that may increase their appeal, and components of the food processing itself (e.g., emulsification) which may yield metabolic consequences [26–28]. The UPF definition is typically applied to packaged foods/food prepared away from home.

Scientific understanding of the availability of both HFP and UPF in the food supply is limited to one prior study of the US food environment over a 30-year period. The study findings revealed that there were different trajectories of the availability of HPF and UPF over time, with UPF increasing 4% from 1988 to 2018, and HPF increasing 14%, indicating their distinctness in the food environment over time. However, by 2018, and the majority (78%) of UPF were also made with combinations of nutrients that classified them as HPF [13]. While this analysis of the US food supply provides some insight into HPF and UPF overlap and distinctness in the US, it is unknown whether patterns of distinctness and overlap are similar or may vary by country, given that there is variability in the degree to which US food companies have a presence in other countries' markets.

Overall, there is a lack of comprehensive research on the availability and characteristics of HPF in the global food supply. Therefore, the purpose of the current study was to 1) compare HPF across countries in a large, globally crowdsourced dataset and examine patterns in availability across HPF groups; 2) compare HPF across countries by the degree to which HPF a) vary by their criteria thresholds across countries, and b) vary by overall nutrient levels across countries; and 3) characterize the degree to which HPF items overlap with UPF or are distinct across countries.

## Methods

### Data source

Data for the current study were obtained from the publicly available Open Foods Facts (OFF) database [29]. OFF hosts a global open-source food database that collects food data contributions from industry, government, and individual contributors globally. OFF has a corresponding mobile app that facilitates individuals in submitting food data on individual foods consumed. The OFF database is designed to be used for research purposes [29] and provides information on food products with detailed nutrition, ingredients, and level of food processing indicated by NOVA score [18]. A NOVA score of 4 identifies ultra-processed foods [18]. Data can be downloaded from OFF and further processed for research purposes (detailed below). OFF data have been used in several research studies in nutritional epidemiology [30–33], illustrating the utility of OFF data for health research. For the current study, a total of 2853365 food items from various countries were downloaded from the OFF database in August 2023. The raw dataset was further processed following the procedures below. The majority of food items included in OFF were packaged or pre-prepared.

### Data processing

Data processing and analyses was conducted using R statistical software [34]. After downloading the raw OFF dataset, quality evaluation procedures were conducted to examine and address data quality concerns before analysis. Specifically, duplicate food items with the same unique identifier (i.e., food code) and identical macronutrient values under the same product name within a country were removed. Food items identified as having a potential data quality issue (e.g., implausible nutrient value; food items contained 0 calories) were excluded. Lastly, food items that were not applicable to the HPF definition (e.g., beverages or items with missing macronutrient values) were removed before analysis. Following quality

control procedures, data were processed to obtain a complete listing of all non-duplicate food items by country. In service of the study focus on HPF, food items in the OFF data that had complete nutrient information for variables needed to apply the HPF definition (total kcal, g per serving, fat, sugar, carbohydrates, fiber, and sodium) were included in analyses. Because the HPF definition does not apply to beverages, beverages were removed from the dataset before analyses. The total number of food items included in the final dataset was 334298. The detailed R code used for the data processing and the data files are publicly available online (https://osf.io/9juwz/).

HPF were identified using the standardized definition of HPF developed by Fazzino and colleagues [5]. This standardized definition was developed using a data-driven approach to identify common nutrient combinations (i.e., fat, carbohydrates, sodium, and sugar) at quantitative thresholds not found in nature [5]. Foods that meet the criteria for at least one of the following groups are identified as HPF: fat and sodium (FSOD, > 25% kcal from fat, >= 0.30% sodium by weight), fat and simple sugars (FS, > 20% kcal from fat, > 20% kcal from sugar), carbohydrates and sodium (CSOD, > 40% kcal from carbohydrates, >= 0.20% sodium by weight).

UPF were identified from the NOVA score provided by OFF, with a score of 4 indicating a food was ultra-processed. NOVA scores were available for 79% of foods in the dataset (Total N = 244863).

The food groups variable in the OFF dataset were utilized to characterize different types of foods. The variable contained information on food categories, which were further specified into main and sub-food categories delimited by commas (e.g., sugary snacks, biscuits, and cakes). Consequently, the food groups variable was split into main and sub-food categories for analysis. Main food categories were mutually exclusive, and each sub-food category belonged to only one main food category. One of the main food categories was "unknown" category, thus food items with unknown main food category was excluded in the study, resulting in 8 main food categories and 31 sub-food categories (S1 Table in S1 File).

## Data coverage and representation across countries

The total number of unique food items varied substantially across countries. Therefore, for analyses, we focused on countries with more than 1000 unique food items. The majority of countries meeting this criterion were from Europe, North America, and South America. Notably, Thailand was the sole Asian country that met this threshold. However, we excluded Thailand from the analysis due to its unique status as the only Asian country; the HPF and UPF definitions were developed for Western food environments and have not been validated with various Asian countries' food environments. The total food items used for the data analysis were 314229 from 17 countries.

The available data were examined for patterns regarding data coverage by country, and potential imbalances in product counts across countries. Regarding data sources, contributions were from governments, industry, individual citizen submissions (through the OFF data reporting app), and commercial apps (see S1 Fig in S2 File for a figure depicting data sources by country). Industry contributions represented approximately half of food items in the data across most countries, with the exception of the US, Poland, Bulgaria, and Mexico, all of which had very limited industry contributions (~10% of data per country). Most countries had limited to no contributions from government sources, whereas the US had the majority of its contributions from government (>70%). Individual user contributions accounted for approximately 20–40% of total items per country, with the exception of Bulgaria, which had > 50% of total products from individual submissions, and the US, which had < 5% of contributions from individual submissions. The degree to which submission sources were not specified accounted for ~10–20% of total food items for most countries, with the exception of Mexico, for which 50% of submission sources were unspecified.

Product counts were examined across countries, to indicate relative size of the available products for analysis, and within countries to examine potential low counts for certain food categories. Regarding total product counts per country, five countries (US, France, Italy, Germany, and Spain) had > 12900 items, four countries had between 5000 and 9700 items, and eight countries (the Netherlands, Poland, Ireland, Portugal, Bulgaria, Australia, Austria, and Mexico) had lower total product counts, ranging from 1174 to 2391 (see Table 1 in the results section). Total product counts did not appear

**Table 1. Food Sample Composition within Country and Food Main Category.**

| Country | N | Percent |
|---|---|---|
| United States | 148841 | 47.37 |
| France | 81470 | 25.93 |
| Italy | 15126 | 4.81 |
| Germany | 14298 | 4.55 |
| Spain | 12982 | 4.13 |
| Switzerland | 9728 | 3.10 |
| Belgium | 7307 | 2.33 |
| United Kingdom | 7296 | 2.32 |
| Canada | 5340 | 1.70 |
| Netherlands | 2391 | 0.76 |
| Poland | 1606 | 0.51 |
| Ireland | 1598 | 0.51 |
| Portugal | 1431 | 0.46 |
| Bulgaria | 1234 | 0.39 |
| Australia | 1230 | 0.39 |
| Austria | 1177 | 0.37 |
| Mexico | 1174 | 0.37 |
| **Food Main Category** | **N** | **Percent** |
| Sugary snacks | 77273 | 24.59 |
| Cereals and potatoes | 70261 | 22.36 |
| Milk and dairy products | 35748 | 11.38 |
| Composite foods | 33879 | 10.78 |
| Fish, Meat, Eggs | 29627 | 9.43 |
| Fats and sauces | 29211 | 9.30 |
| Fruits and vegetables | 22706 | 7.23 |
| Salty snacks | 15524 | 4.94 |

to reflect country or population size, given that some smaller countries had high total product counts (e.g., Italy), whereas some relatively larger countries (e.g., Australia, Mexico) had among the lowest product counts. Regarding representation of products across food categories, seven countries had product counts <100 for at least some food categories, commonly for the categories of fish, meat, and eggs and fruits and vegetables, and these data may be limited in their representativeness of available products in the respective countries.

## Data analysis

To characterize the percentage of HPF in the data by country, we calculated the proportion of HPF overall and the proportion of items per HPF group (fat and sodium HPF; fat and sugar HPF; carbohydrate and sodium HPF), relative to all food items by country. In addition, the proportion of HPF items were identified in each main food category to capture the percentage of HPF by specific type of foods across countries. To examine the differences in the likelihood that food items were classified as HPF across countries, logistic regression was employed. For the logistic regression analyses, the US was used as a reference group to estimate the differences in likelihood food items were classified as HPF. The US was selected as the reference category because the prevalence of HPF in the US food supply has been well documented; HPF saturate the US food supply (68% of available foods) [13,35]. Additional logistic regression models were conducted to determine the likelihood that food items were classified as each HPF group (FSOD, FS, or CSOD).

Second, to understand the differences in nutritional composition of food items classified as HPF across countries, boxplots were produced for each type of HPF, using the quantitative thresholds in the boxplots for each HPF group. Next, we descriptively characterized the variation in the nutrients of food items from HPF criteria to examine the degree to which HPF items generally met or exceeded criteria thresholds. The rate by which nutrients exceeded the HPF criteria thresholds was calculated as the percent by which the nutrient value exceeded the threshold, relative to the threshold value. Finally, ordered beta regression models were employed to examine the statistically significant differences in nutritional compositions of HPF across countries. For this study aim, the outcome variables were percentage variables ranged from 0 to 100, which were continuous with lower and upper bounds. In this case, ordered beta regression is recommended to fit linear regression for both continuous and degenerate (i.e., responses at the lowest and the highest value) outcome [36]. R package "glmmTMB" was used to fit the ordered beta regression model [37].

Finally, to describe the distinct and overlapping prevalence between HPF and UPF across countries, food items were classified into four different categories: a) HPF and UPF; b) HPF but not UPF; c) UPF but not HPF; d) neither HPF nor UPF. Each category represents overlapping HPF and UPF, distinct HPF, distinct UPF, and not HPF and UPF food items. The proportion of each category was identified across countries. A stacked bar plot was used to visualize the distinct and overlapping prevalence in the four categories across countries. Also, to clarify the context of the prevalence between HPF and UPF within the main food categories, stacked bar plots were generated for subsets of food categories across countries. For the analysis of prevalence of HPF and UPF across countries, food items missing NOVA score were excluded. (Food items missing NOVA scores but that contained nutrient information for the standardized definition of HPF were included for the other analyses.)

## Results

### Sample characteristics

As presented in Table 1, the sample consisted of 17 countries and 314229 food items. Food from the US (47.4%) and France (25.9%) were most prominently represented in the data. Across food categories, approximately 50% of total food items were carbohydrate-based sugary snacks or cereals and potatoes. Approximately 30% of foods contained primarily protein and/or fat (fish, meat, eggs and dairy products, fats and sauces) and 7.2% were fruits and vegetables. Although the distribution of food categories varied by country, carbohydrate-based products (the combined proportion of sugary snacks and cereals and potatoes) accounted for at least 40% of the total food items in each country (see S2 Fig in S2 File), with the exception of Bulgaria. Across countries, the variability in available products was higher among food categories such as composite foods, milk and dairy products, salty snacks, and fats and sauces (S1 Table in S1 File). Detailed information on the distribution of food categories across countries is provided in the S1 Table in S1 File.

### Percentage of HPF across countries

Bulgaria had the highest proportion of HPF (67.99%), followed by Switzerland (63.99%), and the US (63.03%). The lowest proportion was observed in Australia (54.47%). The prevalence of food items classified as HPF across countries is reported in S2 Table in S1 File in the Supplemental Information. Regarding HPF types, FSOD comprised the largest percentage of HPF for all countries, followed by FS and CSOD for most countries. For Portugal, Netherlands, and Australia, CSOD were the second most common type of HPF (Fig 1).

Fig 2 presents the prevalence of HPF across each main food category by country. Fig 2 suggested that while the overall prevalence of HPF was consistent across countries, there was noticeable variation in the prevalence of HPF within specific food categories by country. For example, most countries had a very low percentage of fruits and vegetables classified as HPF; however, Bulgaria had approximately 50% HPF in the fruits and vegetables category. There were also some notable consistencies across countries. Approximately 70% of food items in the salty snack category were classified as HPF across all countries. Regarding the composition of HFP groups, FSOD items were predominantly composite

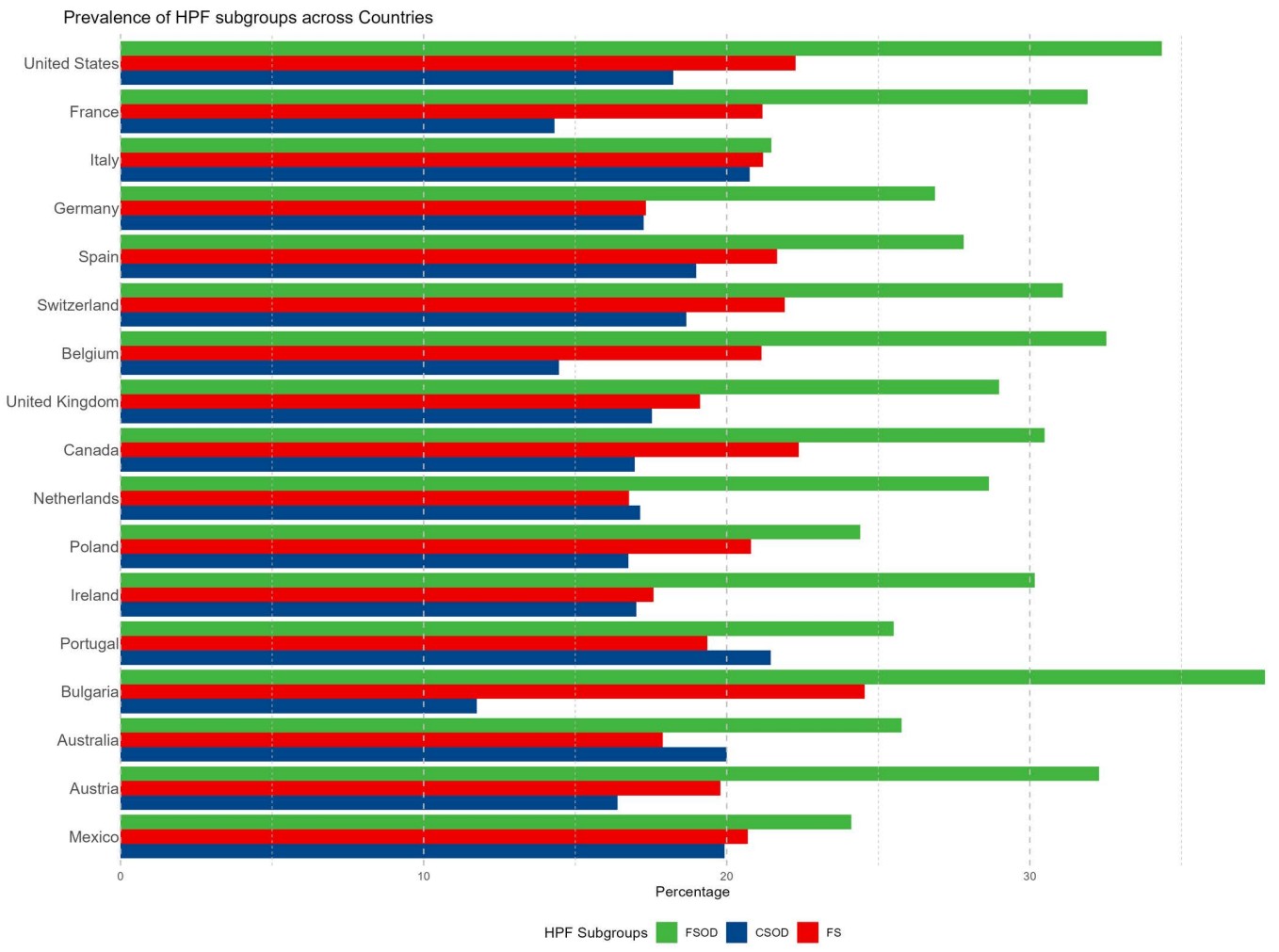

**Fig 1. The Percentage of HPF Groups by Country.**

foods, fats and sauces, salty snacks, and the fish, meat, and eggs categories. CSOD were mostly found in the cereals and potatoes category. FS were primarily found in the sugary snacks category (S3 Fig in S2 File).

The results from the logistic regression model indicated significant differences in the likelihood of food items being classified as HPF across countries, compared to the US (Fig 3 and S3 Table in S1 File). Specifically, most countries had a significantly lower likelihood of their food items being classified as HPF (p-values: <.001 to.013), with odds ratios ranging from 0.70 for Australia to 0.93 for Canada. There was no statistically significant difference between the likelihood of foods being HPF between Switzerland relative to the US. Bulgaria had a significantly higher likelihood of food items being HPF compared to the US (OR = 1.25, p<.001). Overall, the results suggest that foods from the US had an overall higher likelihood of being HPF relative to most other countries examined.

For FSOD HPF, similar findings to the overall HPF model were observed; most countries had a lower likelihood of their food items being classified as FSOD compared to the US, with odds ratios ranging from 0.52 for Italy to 0.92 for Belgium (S4 Table in S1 File and Figure S4 in S2 File). Food items from Austria were not significantly different in the likelihood of being classified as FSOD relative to US items (OR = 0.91, p-value=0.137). On the other hand, Bulgaria had a significantly higher likelihood of food items being classified as FSOD (OR = 1.16, p-value<.012). For FS HPF, ten countries

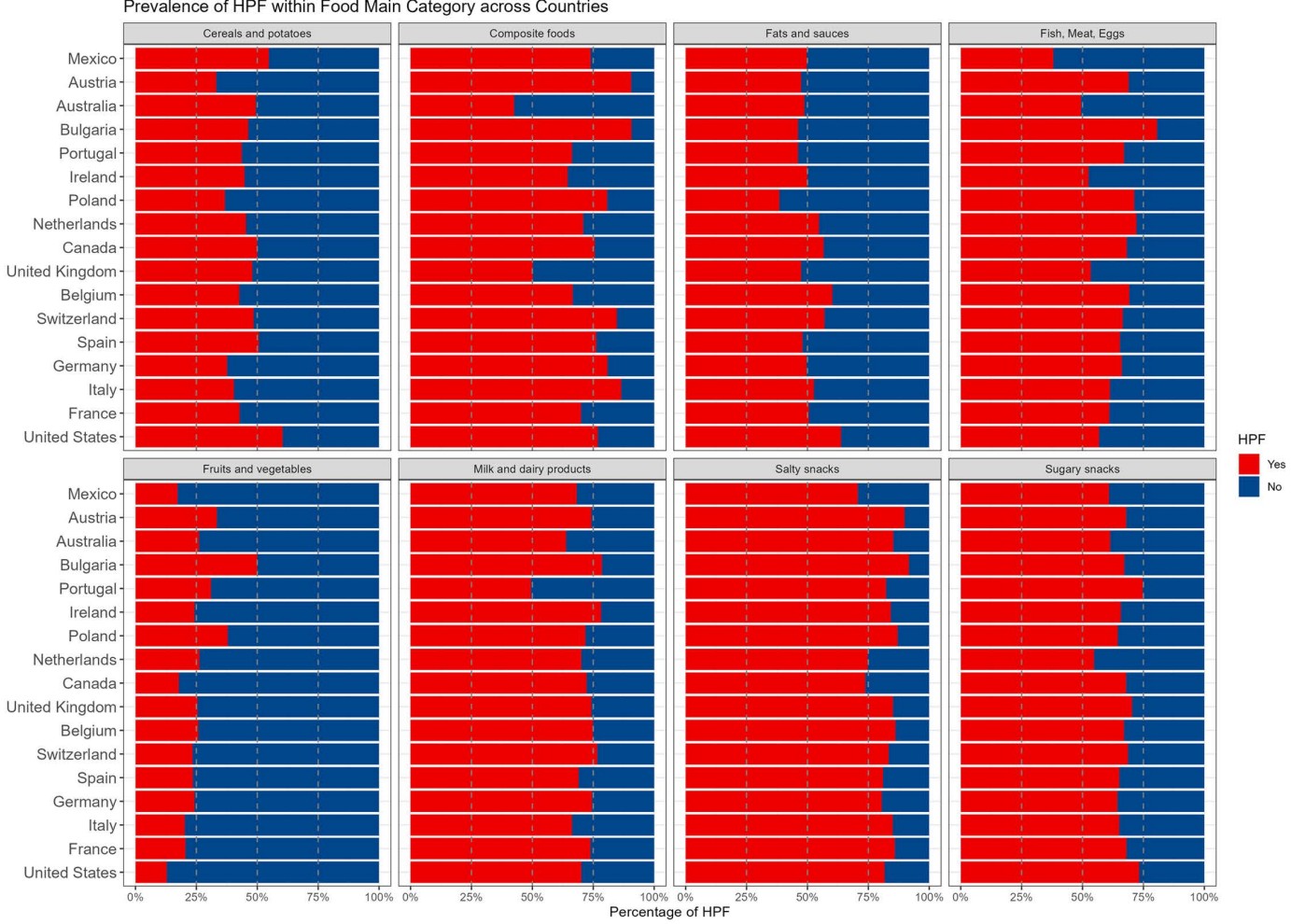

**Fig 2. The Percentage of HPF within food main categories across countries.**

had significantly lower likelihood of food items being FS relative to the US, with odds ratio ranging from 0.73 for Germany to 0.94 for France (S5 Table in S1 File and S5 Fig in S2 File). No significant differences were found between FS from the US and the other seven countries (S5 Table in S1 File and S4 Fig in S2 File). For CSOD HPF, five countries had a significantly lower likelihood of their items being CSOD than the US, with odds ratios ranging from 0.60 for Bulgaria to 0.93 for Germany (S6 Table in S1 File and S6 Fig in S2 File). Italy, Spain, and Portugal had a significantly higher likelihood of food items being CSOD compared to the US (ORs = 1.05 to 1.22). Taken together, the results indicated that foods from most other countries examined had lower likelihood of being classified as FSOD HPF relative to the US; however, the pattern was not consistent with FS and CSOD, highlighting potentially nuanced availability of HPF groups across countries.

## Differences in nutritional composition of HPF across countries

To examine differences in the nutritional composition of foods identified as HPF across countries, descriptive statistics and ordered beta regression models were used. Across countries, the average nutrient values of foods classified as HPF far exceed the HPF criteria thresholds (S7–S9 Tables in S1 File and S7–S9 Figs in S2 File). Specifically, on average across countries, items classified as FSOD exceeded %kcal from fat and % sodium by weight 115.98% and 130.95%

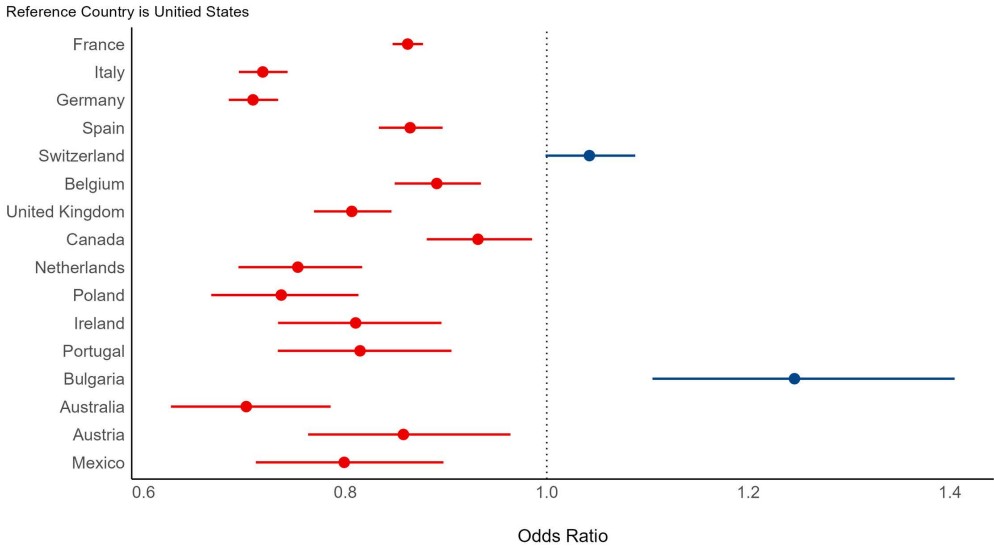

**Fig 3. Odds Ratios of food items being HPF compared to food items from the United States.**

respectively. Items classified as FS surpassed %kcal from fat and % kcal from sugar by 106.04% and 70.15%. Items classified as CSOD exceeded %kcal from carbohydrate and % sodium by weight by 40.04% and 229.25%.

In regression models, results indicated that FSOD items from most countries had significantly lower %kcal from fat than FSOD from the US, with odds ratio ranging from 0.76 for Italy to 0.93 for France. However, FSOD items from Belgium, Canada, and Bulgaria had a significantly higher %kcal from fat relative to FSOD from the US. For % sodium by weight, FSOD items from 7 countries had significantly lower % sodium compared to FSOD from the US, with odds ratio ranging from 0.89 for Portugal to 0.98 for Switzerland (S10 Table in S1 File and S10 Fig in S2 File).

For food items classified as FS, %kcal from fat and % kcal from sugar were compared between the US (reference) and the other countries (S11 Table in S1 File and S11 Fig in S2 File). The results indicated that FS items from most countries had significantly lower %kcal from fat than FS items from the US, with odds ratio ranged from 0.83 for Mexico to 0.97 for Belgium. However, items from Canada and Bulgaria had significantly higher %kcal from fat for FS items relative to US items. Regarding % kcal from sugar, most countries had significantly lower %kcal from sugar in FS items compared to FS items from the US, with odds ratio ranging from 0.83 for Portugal to 0.98 for Switzerland.

Lastly, for food items identified as CSOD, regression results suggested that CSOD items from most countries had significantly lower %kcal from carbohydrates, with odds ratio ranging from 0.85 for Austria to 0.93 for Spain (S12 Table in S1 File and S12 Fig S2 File). On the other hand, CSOD from Canada and Bulgaria had significantly higher %kcal from carbohydrates relative to CSOD from the US, with odds ratio of 1.05 and 1.11 respectively. For % sodium by weight, CSOD items from 10 countries showed significantly lower % sodium compared to CSOD from the US, with odds ratio ranging from 0.80 for the United Kingdom and 0.96 for Switzerland.

### Distinct and overlapping prevalence between HPF and UPF

Fig 4 illustrates the distinct and overlapping prevalence between HPF and UPF. The percent of foods classified as both HFP and UPF ranged from 32.95–50.22% across countries, indicating variability across countries in the degree to which HPF and UPF overlapped (S13 Table in S1 File). Foods classified as distinctly HPF ranged from 13.7–35.5% across

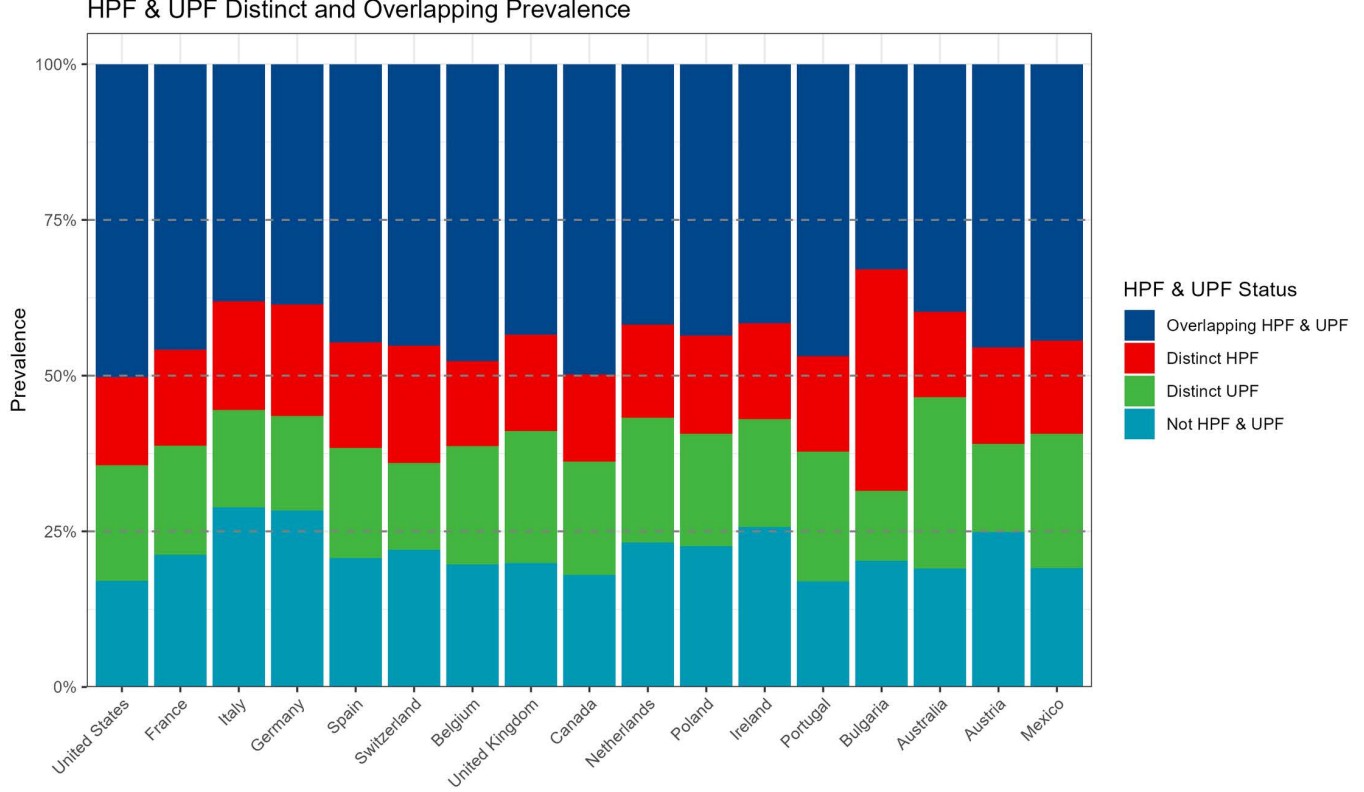

**Fig 4. Distinct and overlapping prevalence between HPF and UPF across countries.**

countries, and foods that were distinctly UPF represented 11.2–27.5% of foods across countries. The US had the highest percentage of food items that met criteria as both HPF and UPF (50.22%) and the lowest percentage of items that were neither HPF nor UPF (17.2%). Bulgaria had the highest percentage of food items classified as distinctly HPF (35.52%). Australia had the highest percentage of food items classified as distinctly UPF (27.49%). Lastly, Italy had the highest percentage of food items not classified as HPF or UPF (28.83%).

Across all countries, food items that were commonly identified as both HPF and UPF were sugary snacks, salty snacks, composite foods, and milk and dairy products (SI S13 Fig in S2 File). Across countries, items that were non-HPF and non-UPF were most commonly in the fruits and vegetables category. In addition, foods that were commonly identified as distinctly HPF (but not UPF) across countries were some salty snacks, milk and dairy products, and fish, meat, and eggs. Foods commonly identified as distinctly UPF (and not HPF) across countries were some sugary snacks, cereals and potatoes, and fats and sauces. Overall, the results suggest patterns of HPF and UPF across countries varied by food categories.

## Discussion

The study used publicly available and globally crowdsourced data to examine characteristics of foods across countries. Using data available from 17 countries across Europe, North America, and South America, the study was the first to examine and compare categories of foods that have been associated with overeating and obesity in the literature, specifically hyper-palatable foods and ultra-processed foods. The data source provided >300000 unique foods for analyses, which facilitated examinations of patterns across countries and food types. Findings indicated that foods from most countries

from Europe, as well as Canada and Mexico, had significantly lower likelihood of being hyper-palatable, relative to US food items. Furthermore, results indicated that HPF from most countries had significantly lower percent calories from fat, sugar, starchy carbohydrates, and percent sodium (in grams) relative to HPF from the US, although Bulgaria and Canada were exceptions. Finally, analyses indicated variability in the degree to which food items were classified as HPF, UPF, or both across countries; the US had the highest percentage of food items classified as HPF and UPF, whereas Bulgaria had the lowest percentage of items that were both HPF and UPF. Overall, analyses of a rich, publicly sourced dataset identified several patterns across food types and their nutritional characteristics across countries in Europe, North America, and South America, generally identifying US foods as more likely to be HPF and contain higher nutrient levels (e.g., fat and sugar) relative to foods from countries in Europe, as well as Canada and Mexico.

The Open Food Facts dataset used in the study provided a wealth of food data across countries globally; however due to the crowdsourced nature of the data, estimates of HPF availability by country could not be considered representative of a country's food environment. To this point, there was wide variability in the total products available for analysis by country, and approximately half of countries analyzed had lower product counts in at least one food category, suggesting limited representation for at least some food products. Due to these limitations in representation of country-level food environments, the OFF data were examined to glean overall patterns and to examine HPF by type. Findings generally indicated that US food items were more likely to be HPF relative to most other countries examined across Europe, Canada, and Mexico, and the pattern was remarkably consistent across most countries. There were two exceptions; food items from Switzerland were not significantly different in their likelihood of being classified as HPF relative to US items, and items from Bulgaria were significantly more likely to be HPF relative to US items. Regarding countries with healthier food profiles, Italy and Germany had food items that had a substantially lower likelihood of being classified as HPF relative to US items in analyses. Although these analyses are relatively unique to the literature, they are consistent with the findings of one prior study that compared the US and Italian food environments and found that the Italian food environment had substantially lower HPF availability relative to the US for the same year [38]. A similar pattern to the main HPF findings emerged when examining HPF by group; for fat and sodium HPF, fourteen countries had significantly lower likelihood of their food items being fat and sodium HPF relative to US items, with effect estimates ranging from 8–48% lower. The exception was food from Bulgaria, which had a greater likelihood of being fat and sodium HPF relative to US items. Food items from Italy were the least likely to be fat and sodium HPF relative to the US and all other countries examined. The pattern was less consistent for fat and sugar HPF and carbohydrate and sodium HPF, however US items had higher likelihood of being identified as fat and sugar or carbohydrate and sodium relative to ~30–59% of countries (5–10 countries out of 17). Food items from Germany were the least likely to be fat and sugar HPF and food items from Bulgaria were least likely to be carbohydrate and sodium HPF. Findings overall suggest that while many countries may have their share of HPF, the US stands out as having comparatively higher HPF availability. The findings also reveal that foods from some countries, primarily Italy and Germany, were substantially less likely to be HPF when examined as HPF overall and by HPF group, which may be more protective against excess intake for their populations.

Another pattern observed across countries was the distribution of HPF types; across countries, fat and sodium HPF comprised the greatest percentage of HPF for all countries, followed by FS and CSOD for most countries. This pattern is highly consistent with prior findings from studies that used representative food environment data from the US; fat and sodium HPF comprised the largest percentage of available HPF, and this distribution in the food supply has been remarkably consistent since the late 1980's [13,35]. Findings from the Italian food environment also support this premise; fat and sodium HPF comprised ~60% of available HPF in the Italian food supply [38]. Fat and sodium HPF typically consist of meal-based items in the US environment (e.g., meat dishes, combination meat and grain dishes, etc) and specialty meat and cheese products in the Italian food environment, which may partially explain their overall prevalence among HPF items across countries examined in the current study. Taken together, findings from the current study and the prior

literature support the premise that fat and sodium HPF appear to be the most common type of HPF among countries examined. The findings also align with our examination of HPF across food types, such that fat and sodium HPF mostly comprised composite foods, fats and sauces, and fish, meat, eggs food categories.

The study was the first to conduct comparisons of products that met criteria for each HPF group across countries, to understand the degree to which HPF formulations may vary by country. Overall, findings indicated that on average, HPF items across all countries substantially exceed HPF criteria thresholds. Most HPF items had nutrient values that were >100% of the HPF threshold criteria, suggesting that on average most foods were formulated with nutrient levels well beyond the thresholds for HPF using the standardized definition. When examining the nutrient levels of HPF items across countries, analyses indicated that HPF produced in the US contained on average significantly higher %kcal from fat, %kcal from sugar, %kcal from carbohydrates, and sodium relative to HPF produced in most European countries and Mexico. However, a pattern emerged in which foods from Bulgaria and Canada had significantly higher % calories from fat among fat and sodium HPF items and fat and sugar HPF items, and significantly higher % calories from carbohydrates among carbohydrate and sodium HPF. Bulgaria was less well represented among the study data, particularly in some food categories (e.g., composite foods, salty snacks), and future work should seek to replicate these findings. Future research should also examine HPF in Canada, given that Canadian foods were better represented in the data and appeared to have HPF item nutrients that exceed averages even for US HPF items.

A final focus of the study was in examining the overlap and distinction of HPF with UPF across countries. Both HPF and UPF have distinct paths by which they may influence overeating and obesity consequences, and foods that are categorized as both HPF and UPF may be particularly concerning for public health. In contrast, foods that are categorized as neither HPF nor UPF, which are most commonly whole, fresh fruits and vegetables, may be considered particularly health promoting. Findings overall revealed substantial differences in the degree to which foods across countries were classified as overlapping or distinct in their categorizations. The US had the highest percentage of foods identified as both HPF and UPF (50%) and the smallest percentage of foods that were neither HFP nor UPF (17%). The findings are highly consistent with prior analyses of US food system data for 2018, in which a small minority of foods (<15%) in the total food supply were neither HPF nor UPF [13]. In contrast to the US, Bulgaria had the lowest percentage of foods classified as both HPF and UPF (32.9%), followed by a high percentage of foods that were HPF only (35.5%), and the smallest percentage of items classified as UPF only (11%) across all countries. Thus, Bulgaria appears to represent an environment in which HPF are largely distinct from UPF products; however given the limited data availability from Bulgaria relative to most other countries analyzed, replication of these findings should be examined future work. Italy had among the lowest percentages of foods that were both HPF and UPF (38%) and had the highest percentage of foods that were neither HFP nor UPF (28.8%), suggesting that the Italian food environment may be relatively more health-promoting with non-HPF/UPF food options, which is consistent with one prior analysis of representative food environment data [38]. Overall findings suggested that when examining data outside of the US food system, there may be greater distinction across countries in the degree to which foods may be HPF or UPF, which supports the premise that the two are not synonymous with each other. Given that the US may have a saturated food environment due to the presence and origin of several multi-national food companies that largely produce HPF and UPF, the degree to which these foods have entered markets globally varies. Therefore, our findings suggest that examining HPF and UPF separately across countries food environments is necessary to understand varied risks that each may present for overeating and obesity and chronic disease risk.

Additionally, despite the well-documented health consequences of high UPF intake [39], the conversation in the literature regarding UPF has become more nuanced and has begun asking whether all UPF may be equally detrimental to health [40–42], and whether some UPF may even contribute to higher dietary quality [43]. Our findings suggest that a substantial percentage of foods in most countries' food environments that we examined have foods that are both UPF and HPF; given that HPF are mechanistically related to the reinforcing properties of a food [44], UPF that are also HPF may be

particularly hard to stop eating, and risky for population health. Foods that are UPF but not HPF may be less concerning regarding their reinforcing properties and risk for overeating; however UPF (that are not HPF) may contain other additives and undergo specific types of processing (e.g., emulsification) that may be detrimental to metabolic health [27]. Ongoing research is working to examine which aspects of UPF may yield their negative health outcomes [45]. Our work overall indicates that several types of UPF (UPF that are hyper-palatable and those that are not) are available in most countries' food environments and may represent a risk to population health.

The study had several limitations. First, the data analyzed were crowdsourced data and should not be considered representative of country-level food environments. Because of this limitation, we focused on analyzing patterns across countries and in making comparisons across food characteristics (but not prevalences) across countries. In addition, the representation of countries in the dataset was largely skewed toward Western Europe and North America, leaving out entire continents of Africa and Asia, as well as many other countries globally. Although the HPF definition was developed on and primarily applicable to Western food environments, we recognize the limitations in the representativeness of the dataset used in analyses. However, one benefit of Open Food Facts is that individuals and agencies globally can contribute data; therefore, one approach could be to bolster reporting by countries in an effort to increase data flow and representation in Open Food Facts, which could support future work in this area.

Strengths of the study include the use of a large and detailed food nutrient dataset with contributors globally, and the nuanced analyses that were the first to examine characteristics of HPF among 17 countries in Europe, North America, and South America.

## Conclusions and implications

The study leveraged a globally crowdsourced data source to examine patterns and nutrient characteristics of hyper-palatable foods across countries and their overlap and distinction with ultra-processed foods. Findings across 17 countries across Europe, North America, and South America highlighted foods from the US as being more likely to be hyper-palatable relative to most other countries examined, particularly HPF that have elevated fat and sodium. Furthermore, a direct comparison of the nutrient characteristics of HPF across countries revealed that HPF produced in the US had on average significantly greater calories from fat, sugar, starchy carbohydrates, and sodium relative to HPF from most other countries examined. The findings indicate that not all HPF are created equally, and HPF produced in the US appear to have greater levels of palatability-related nutrients relative to most other countries. In contrast, foods from Italy and Germany were consistently identified across analyses as having substantially lower likelihood of being HPF relative to the US, suggesting both countries may have more health-promoting non-HPF options in their food environments. Finally, from the current study highlighted variability in the degree to which foods across countries meet criteria for HPF and/or UPF, further supporting the premise that HPF and UPF should be examined separately across countries, given that they may have different paths contributing to excess energy intake and chronic disease risk.

Findings of the study largely indicate that HPF are prominent across many countries. Obesogenic food environments that may be particularly challenging to population health are those in which HPF are widely available, easily accessible, and inexpensive to purchase. The food environment has been long established as a systemic driver of food availability and obesity risk [46,47] and requires a system-level approach to best protect population health. Countries may consider public policy regulation to limit the presence of HPF in their food environments through limitations on sales, locations in stores (e.g., moving items from prominent positions at front of store check out/payment station to the back of a store), and age restrictions on who can purchase HPF (e.g., adults). Simultaneously, countries could encourage or require the reformulation of HPF products' nutrients to be below the threshold criteria for HPF, to reduce the number of HPF items in the broader food supply. For example, for a pre-packaged meal that is fat and sodium HPF, food companies could be required to reduce the sodium content below the HPF sodium threshold criterion, which would render the meals as not HPF, which then could be sold openly (without restrictions) to the public in food stores. This approach would allow for the retention of

all food products in the food supply, while reducing the reinforcing properties of HPF that may drive overconsumption and risk population health.

## Supporting information

**S1 File.** **S1 Table**. Prevalence of HPF within Food Main Categories across Countries. **S2 Table**. Summary of the Prevalence of HPF across Countries.**S3 Table**. Logistic regression results for the HPF prevalence compared to United States. **S4 Table**. Logistic regression results for the FSOD prevalence compared to United States. **S5 Table**. Logistic regression results for the FS prevalence compared to United States. **S6 Table**. Logistic regression results for the CSOD prevalence compared to United States. **S7 Table**. Descriptive Statistics of Nutritional Compositions of FSOD across Countries.**S8 Table**. Descriptive Statistics of Nutritional Compositions of FS across Countries. **S9 Table**. Descriptive Statistics of Nutritional Compositions of CSOD across Countries. **S10 Table**. Ordered Beta Regression Results for nutritional compositions of FSOD across countries compared to United States.**S11 Table**. Ordered Beta Regression Results for nutritional compositions of FS across countries compared to United States. **S12 Table**. Ordered Beta Regression Results for nutritional compositions of CSOD across countries compared to United States. **S13 Table**. Distinct and Overlapping percentage of HPF and UPF across countries.
(ZIP)

**S2 File.** **S1 Fig**. Data Source Composition by Country. **S2 Fig**. The Proportion of Main Food Category within Each Sampled Country. **S3 Fig**. Prevalence of HPF groups within Food Main Categories across Countries. **S4 Fig**. 95% Confidence Interval for Odds Ratio of food items being FSOD compared to United States. **S5 Fig**. 95% Confidence Interval for Odds Ratio of food items being FS compared to United States.**S6 Fig**. 95% Confidence Interval for Odds Ratio of food items being CSOD compared to United States.**S7 Fig**. Boxplot for the nutritional compositions of FSOD across countries.**S8 Fig**. Boxplot for the nutritional compositions of FS across countries.**S9 Fig**. Boxplot for the nutritional compositions of CSOD across countries. **S10 Fig**. 95% Confidence intervals plot for nutritional compositions of FSOD compared to the United States.**S11 Fig**. Confidence intervals plot for nutritional compositions of FS compared to the United States.**S12 Fig**. Confidence intervals plot for nutritional compositions of CSOD compared to the United States. **S13 Fig**. Distinct and overlapping prevalence between HPF and UPF across countries within food main categories.
(ZIP)

## Author contributions

**Conceptualization:** Daiil Jun, Kelly Knowles, Tera L. Fazzino.

**Data curation:** Daiil Jun, Kelly Knowles.

**Formal analysis:** Daiil Jun, Kelly Knowles.

**Funding acquisition:** Kelly Knowles.

**Investigation:** Tera L. Fazzino.

**Methodology:** Daiil Jun, Kelly Knowles, Tera L. Fazzino.

**Project administration:** Daiil Jun.

**Supervision:** Tera L. Fazzino.

**Validation:** Daiil Jun.

**Visualization:** Daiil Jun, Kelly Knowles.

**Writing – original draft:** Daiil Jun, Kelly Knowles.

**Writing – review & editing:** Tera L. Fazzino.

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
