## [Decision Letter · Decision Letter 0]

23 Jan 2025

Dear Dr. Fazzino,

Thank you for submitting your manuscript to PLOS ONE. After careful consideration, we feel that it has merit but does not fully meet PLOS ONE’s publication criteria as it currently stands. Therefore, we invite you to submit a revised version of the manuscript that addresses the points raised during the review process.

Please note that we have only been able to secure a single reviewer to assess your manuscript. We are issuing a decision on your manuscript at this point to prevent further delays in the evaluation of your manuscript. Please be aware that the editor who handles your revised manuscript might find it necessary to invite additional reviewers to assess this work once the revised manuscript is submitted. However, we will aim to proceed on the basis of this single review if possible. 

We look forward to receiving your revised manuscript.

Kind regards,

Joanna Tindall, PhD

Staff Editor

PLOS ONE

“Grants from the Kansas Idea Network of Biomedical Research Excellence (P20 GM103418; PI: Wright) and the University of Kansas Center for Undergraduate Research supported author KK’s time during the study.”

Reviewers' comments:

Reviewer's Responses to Questions

**Comments to the Author**

1. Is the manuscript technically sound, and do the data support the conclusions?

Reviewer #1: Yes

2. Has the statistical analysis been performed appropriately and rigorously?

Reviewer #1: Yes

3. Have the authors made all data underlying the findings in their manuscript fully available?

Reviewer #1: Yes

4. Is the manuscript presented in an intelligible fashion and written in standard English?

Reviewer #1: Yes

Reviewer #1: I would like to thank the authors of this manuscript. Having read your manuscript, I understand that it investigates the prevalence and nutrient characteristics of hyper-palatable foods (HPF) across 17 countries, examining their potential link to increased energy intake and obesity. The authors utilize crowdsourced data from the Open Food Facts database to compare the availability of HPF across different countries and analyze their nutrient composition. Additionally, the study explores the overlap between HPF and ultra-processed foods. I have a few suggestions that may further enhance your manuscript:

Abstract

1.1 The abstract is concise and informative. It clearly states the study’s purpose, methodology, key findings, and implications. However, based on the information the authors have provided, it seems that the comparison between HPF and UPF is a significant part of the research. Therefore, it would be beneficial to briefly mention UPF in the abstract to highlight the comparison between HPF and UPF as a key aspect of the research.

Introduction

2.1 While the introduction provides a good overview of HPF and their association with obesity risk, it could be strengthened by expanding on the global prevalence of HPF and the need for more research in this area.

2.2 The authors should explicitly state the research gap that this study aims to fill, such as the lack of comprehensive research on the global prevalence and characteristics of HPF.

Methods

Please provide additional information about the Open Food Facts database is essential for transparency and replicability, including details on data collection and quality assurance procedures.

Discussion

4.1 Please define strengths of your study

4.2 A deeper discussion of UPF, relating it back to the specific findings of the study and highlighting the prevalence of foods classified as both HPF and UPF, would enhance the analysis.

4.3 The authors should compare the study’s findings with previous research, highlighting any similarities or discrepancies and discussing potential explanations for any differences.

4.4 It would be helpful to discuss the implications of the findings for public health interventions.

**Do you want your identity to be public for this peer review?** For information about this choice, including consent withdrawal, please see our Privacy Policy

Reviewer #1: No

---

## [Author Response · Author response to Decision Letter 0]

29 Jan 2025

Please see response to reviewers document (attached)

---

## [Decision Letter · Decision Letter 1]

25 Apr 2025

Dear Dr. Fazzino,

Thank you for submitting your manuscript to PLOS ONE. After careful consideration, we feel that it has merit but does not fully meet PLOS ONE’s publication criteria as it currently stands. Therefore, we invite you to submit a revised version of the manuscript that addresses the points raised during the review process.

We look forward to receiving your revised manuscript.

Kind regards,

John W. Apolzan, PhD

Academic Editor

PLOS ONE

Journal Requirements:

Additional Editor Comments:

Please note few comments that remain to be addressed.

Could you please review the comments and address them accordingly. 

Reviewers' comments:

Reviewer's Responses to Questions

**Comments to the Author**

Reviewer #1: All comments have been addressed

Reviewer #2: (No Response)

2. Is the manuscript technically sound, and do the data support the conclusions?

Reviewer #1: Yes

Reviewer #2: Partly

3. Has the statistical analysis been performed appropriately and rigorously?

Reviewer #1: Yes

Reviewer #2: Yes

4. Have the authors made all data underlying the findings in their manuscript fully available?

Reviewer #1: Yes

Reviewer #2: Yes

5. Is the manuscript presented in an intelligible fashion and written in standard English?

Reviewer #1: Yes

Reviewer #2: Yes

Reviewer #1: (No Response)

Reviewer #2: This manuscript presents a well-designed and methodologically transparent cross-national analysis of hyper-palatable foods (HPF) using a large crowdsourced dataset (Open Food Facts). The study applies a validated operational definition of HPF and contributes original insights into cross-country differences in food formulation and the overlap with ultra-processed foods (UPF). Statistical methods are appropriate and clearly reported, and the discussion is well-aligned with the findings.

Nonetheless, a few aspects could be strengthened to improve clarity and interpretability. First, the conceptual distinction between HPF and UPF—though central to the manuscript—would benefit from clearer explanation earlier in the introduction or discussion. This would help readers unfamiliar with the field better contextualize the comparative findings. Second, while the dataset is extensive, the representativeness of country-level comparisons remains limited. A more detailed discussion of data coverage across countries (e.g., imbalances in product counts, potential input bias, or contributor type) would help interpret the generalizability of results.

Although the manuscript acknowledges the limitations of working with a crowdsourced database, highlighting a few critical patterns from the supplemental figures in the main text—particularly for nutrient composition—could improve synthesis. The analysis of HPF–UPF overlap is a novel strength of the study and could be more prominently presented in the Results section. Finally, while the manuscript is written in standard English, minor stylistic improvements in phrasing and paragraph transitions could enhance readability, particularly in the Results and Discussion sections.

In summary, this is a relevant and technically sound contribution to the literature on food system environments and public health nutrition. I recommend publication pending minor to moderate revisions, focused on interpretive framing and clarity of presentation.

**Do you want your identity to be public for this peer review?** For information about this choice, including consent withdrawal, please see our Privacy Policy

Reviewer #1: No

Reviewer #2: **Yes: ** Sinara Laurini Rossato

---

## [Editor Report · Decision Letter 2]

14 May 2025

Examination of Hyper-Palatable Foods and their Nutrient Characteristics using Globally Crowdsourced Data

PONE-D-24-53803R2

Dear Dr. Fazzino,

We’re pleased to inform you that your manuscript has been judged scientifically suitable for publication and will be formally accepted for publication once it meets all outstanding technical requirements.

Kind regards,

John W. Apolzan, PhD

Academic Editor

PLOS ONE

---

## [Editor Report · Acceptance letter]

PONE-D-24-53803R2

PLOS ONE

Dear Dr. Fazzino,

I'm pleased to inform you that your manuscript has been deemed suitable for publication in PLOS ONE. Congratulations! Your manuscript is now being handed over to our production team.

Kind regards,

on behalf of

Dr. John W. Apolzan

Academic Editor

PLOS ONE